# *Vibrio cholerae* Bacteremia: An Enigma in Cholera-Endemic African Countries

**DOI:** 10.3390/tropicalmed9050103

**Published:** 2024-05-02

**Authors:** Foster K. Agyei, Birgit Scharf, Samuel Duodu

**Affiliations:** 1West African Centre for Cell Biology of Infectious Pathogens, University of Ghana, Accra LG54, Ghana; fosterka@vt.edu; 2Department of Biological Sciences, Virginia Polytechnic Institute and State University, Blacksburg, VA 24061, USA; bscharf@vt.edu; 3Department of Biochemistry, Cell and Molecular Biology, University of Ghana, Accra LG54, Ghana

**Keywords:** *Vibrio cholerae*, non-O1/non-O139 bacteremia, sub-Saharan Africa, virulence factors, diagnosis

## Abstract

Cholera is highly endemic in many sub-Saharan African countries. The bacterium *Vibrio cholerae* is responsible for this severe dehydrating diarrheal disease that accounts for over 100,000 deaths each year globally. In recent years, the pathogen has been found to invade intestinal layers and translocate into the bloodstream of humans. The non-toxigenic strains of *V*. *cholerae* (non-O1/O139), also known as NOVC, which do not cause epidemic or pandemic cases of cholera, are the major culprits of *V*. *cholerae* bacteremia. In non-cholera-endemic regions, clinical reports on NOVC infection have been noted over the past few decades, particularly in Europe and America. Although low–middle-income countries are most susceptible to cholera infections because of challenges with access to clean water and inappropriate sanitation issues, just a few cases of *V*. *cholerae* bloodstream infections have been reported. The lack of evidence-based research and surveillance of *V. cholerae* bacteremia in Africa may have significant clinical implications. This commentary summarizes the existing knowledge on the host risk factors, pathogenesis, and diagnostics of NOVC bacteremia.

## 1. Introduction

Cholera is a severe diarrheal disease caused by the *V. cholerae* bacterium, a prototypical non-invasive mucosal pathogen. It is a major reason of fatal infections in many underdeveloped regions with inadequate access to clean drinking water and remains an important global health problem [1]. The toxigenic strains of the O1 and O139 *V*. *cholerae* serogroups, which bear the cholera toxin (CTX), are implicated in all cholera outbreaks and pandemic spread. On the other hand, non-toxigenic strains that generally lack CTX production, namely the non-O1 and non-O139 *V*. *cholerae* (NOVC), usually do not cause severe diarrhea as seen with choleragenic vibrios [2]. However, NOVC are known to cause bacteremia and other extraintestinal infections including those affecting the skin, wounds, the biliary or urinary tract, and the brain [3]. Between 2009 and 2014, a statistical analysis of 83 NOV-infected patients in Taiwan revealed that gastroenteritis was the most prevalent NOVC-related illness (54.2%), followed by biliary tract infection (14.5%), and primary bacteremia (13.3%) [4]. Despite being uncommon, NOVC bacteremia has the greatest fatality rate of up to 39% [4,5]. Between 1980 and 2014, 175 cases of NOVC bacteremia were reported with high incidence rates in Asia (59%), followed by Europe (14%), United States (11%), and other countries (15%) [6]. The number of cases found doubled by extending the literature search to include other publications from 1974 to 2014 [7]. Most of these cases (45%) originated in Taiwan, 20% in the United States, and 6% in Spain. Although the number of cases has risen worldwide in recent years [5,8,9], only four documented reports of *V. cholerae* bacteremia have been confirmed in Africa, specifically in Malawi, Mauritius, and South Africa [10,11,12,13]. Four other reports could be traced to African origin but diagnosed and/or confirmed in Europe [7,14,15,16]. These patients reported to the hospital with varying forms of clinical symptoms such as hypoglycemia, hypothermia, fever, tachycardia, and random episodes of watery or bloody diarrhea. NOVC strains were mostly isolated from these African patients. The three cases of *V*. *cholerae* bacteremia from Malawi attributed to toxigenic O1 strains occurred following a community cholera outbreak. Among these O1 bacteremia patients were neonates, who did not present with diarrhea symptoms, but died because of the infection probably from a cholera infected asymptomatic mother [10]. Nevertheless, the rare occurrence of *V. cholerae* bacteremia in Africa is surprising as cholera is endemic in many African countries and high numbers of *V. cholerae* bacteremic cases have been reported in Asia and the Middle East, where cholera disease is also widespread (Table 1, Figure 1). It is not well established if the rare occurrence of reported *V. cholerae* bacteremia is a true reflection of the epidemiological situation in Africa. 

NOVC are genetically diverse strains with a wide ecological niche [17]. They mostly inhabit estuarine and marine environments. Several NOVC strains have been isolated from the coastal waters in Europe (Germany, France) and Asia (India and South Korea) [18,19,20]. Humans can contract infections through eating raw or undercooked meat and seafood, or through skin lesions exposed to contaminated seawater [7]. Seafoods contributing to NOVC bacteremia include oysters, fish, shrimps, clams, and mussels. Non-O1/non-O139 infections have also been demonstrated to be caused by exposure through human swimming activities in freshwater lakes. Several emerging reports have linked NOVC to inland water sources including ponds and other freshwater bodies across the globe [8,21,22,23]. Despite all these reports, the epidemiology of NOVC bacteremia has yet to be clarified. In more than 75% of patients, the infection source cannot be determined. Climatic conditions such as an increase in sea surface temperatures may exacerbate the ecological risk for NOVC transmission and infection. For instance, the heat waves that swept through the coastal areas of Sweden and Finland in the summer of 2014 coincided with an increased number of human NOVC infections, which even extended far north into the subarctic regions [24]. There is a strong conviction that NOVC infections are becoming more frequent in the northern hemisphere due to climate change [25]. However, the impact of climate change is not peculiar to temperate and Mediterranean regions. Most countries in sub-Saharan Africa are also experiencing erratic climatic conditions, including heavy rainfall and prolonged drought, that increases the chances of *V*. *cholerae* infections. It is important to note that NOVCs were in existence prior to the discovery of pathogenic *Vibrio* strains [26]. The reason for neglected attention to NOVC bacteria might be because most people believed that they are non-pathogenic colonizers in humans and not associated with deadly disease outcomes. However, considering the clinical significance of NOVC bacteremia, more consideration should be given to the study of this disease. This commentary addresses some key aspects of the clinical presentations, host risk factors, pathogenesis, diagnosis, and antibiotic therapy of NOVC bacteremia. 

**Table 1 tropicalmed-09-00103-t001:** Reported cases of *Vibrio cholerae* bacteremia in 5-year intervals between the years 2000–2023.

Year	Countries	Number of Cases	References
2000–2005	Australia, Denmark, Spain, USA, Kuwait, Thailand, Malawi, Belgium, India, Korea, France, Taiwan, Slovenia	38	[10,14,16,27,28,29,30,31,32,33,34,35,36,37,38,39,40,41,42,43,44]
2006–2011	USA, China, Finland, Poland, India, Taiwan, Austria, France, Qatar, Thailand, Chile, Malaysia, Italy, France, Argentina, Lebanon	66	[11,15,45,46,47,48,49,50,51,52,53,54,55,56,57,58,59,60,61,62,63,64,65,66,67]
2012–2017	Japan, China, Argentina, Thailand, Australia, UK, India, Saudi Arabia, New Zealand, Portugal, Vietnam, France, Netherlands, Austria, Greece, Pakistan, Canada	47	[2,3,6,7,68,69,70,71,72,73,74,75,76,77,78,79,80,81,82,83,84,85,86,87,88,89]
2018–2023	Pakistan, China, Belgium, Lebanon, USA, Saudi Arabia, South Africa, India, Oman, Colombia	34	[5,8,9,13,90,91,92,93,94,95,96,97,98,99,100,101,102,103,104]

## 2. NOVC Clinical Presentation and Host Risk Factors

While NOVC bacteremia presents with a wide number of clinical manifestations; fever, epigastric discomfort, chills, low blood pressure, nausea, overall weakness, occasional vomiting, dizziness, mild diarrhea, and sensations of lethargy were among the common clinical symptoms reported to the hospital from a clinical study [5]. Infected persons usually do not present the typical profuse watery diarrhea symptoms as in the case of cholera [80]. Aside from an increased pulse rate (about 150 beats/min), the patient may have abnormal changes in hematological and inflammatory parameters [8]. Increased white blood cell count, neutrophils, and elevated levels of C-reactive protein have been documented [8,17]. The lower limb pains experienced by some patients may be a result of inflammation. Although NOVC has been isolated from asymptomatic human carriers [3], the disease primarily affects immunocompromised individuals and patients with chronic illnesses [58]. Specifically, chronic infections involving the liver or spleen raise a potential risk as these organs oversee the removal of active bacteria from the bloodstream. This concept is well supported by the fact that many of the affected patients had underlying illnesses related to liver disorders. The susceptibility of cirrhotic liver patients to NOVC bacteremia is believed to be associated with intestinal mucosal inflammation and edema, as well as heightened intestinal permeability [7]. Alcoholism, diabetes mellitus, hematological malignancies, cancer, and steroid use (prolonged corticosteroid therapy) are additional risk factors associated with NOVC bacteremia [7]. Diabetic patients are often in a state of persistent hyperglycemia that causes the dysfunction of the immune system, exposing the individual’s body to attack from not just NOVC strains, but other pathogens in general [105]. Individuals on antacids or undergoing immune-suppressing therapy are also more prone to NOVC and likely to develop more progressive infections [106]. It has been reported that men are more susceptible to NOVC infection than women. Middle-aged men (average age of 56 years and above) are at high risk for NOVC infection compared to children under 18 years [5,7], suggesting that an aging demographic may increase the likelihood of the infection. Compared to Europe and the United States, Africa has a youthful population. Nearly two-thirds of the population in sub-Saharan Africa is under 25 years old, with only 3 percent within the age bracket of 65 years and above. Contrary to what is seen in most developed countries with a much older population (median age between 38 and 43 years), fewer people in Africa are expected to have underlying predisposing conditions that can increase the risk of developing severe disease such as those mediated by NOVC [107]. If this correlation is true, why are countries in South and Southeast Asia reporting more cases of *V*. *cholerae* bacteremia despite having a young population (with a median age of 28), just as in Africa? 

## 3. Pathogenesis of NOVC Bacteremia

The causative mechanisms of NOVC bacteremia are not well understood. However, it is thought to spread from the small intestine or skin wounds into the blood–lymphatic system. The virulence features of NOVC strains present exceedingly serological variability. Despite the absence of numerous key virulence factors such as the CTX and toxin-coregulated pilus (TCP) in non-O1/non-O139 serogroups [108], several synergistic factors play significant roles in the infection process. The toxin-regulating gene product (ToxR), repeats-in-toxin (RTX), hemolysins (HlyA), hemagglutinin protease (HAP), type III (T3SS) and type VI (T6SS) secretion systems, previously found in toxigenic strains, are among the predominant accessory virulence factors identified in NOVC strains that are associated with pathogenicity [109,110]. The potential contribution of these virulence factors to NOVC bloodstream invasion has also been noted from several studies, based on enhanced hemolytic and/or cytotoxic properties of the strains and ability to induce cell vacuolation [45,80,104,111]. The non-O1/non-O139 type-three secretion system (T3SS) exhibits homology to the *V. parahaemolyticus* T3SS gene clusters, postulated to play a role in virulence and environmental adaptation [110]. Knowing that *V. parahaemolyticus* also causes bacteremia just like NOVC, suggests parallel mechanisms in their pathogenesis. Acid neutralization, antiphagocytosis, iron acquisition, cytotoxicity, attachment, adhesion, flagella, and motility are some essential virulence features shared between these two pathogens [112,113]. Notably, a significant number of NOVC isolates are characterized by the presence of capsules, which may be critical for virulence in bacteremia infections [104,114]. While the role of most virulent factors has been well studied, the function of targeted, flagellar-driven motility is poorly understood. Chemotaxis in *V. cholerae* is complex, organized in three chemosensory systems, using 43 chemoreceptor proteins to sense environmental compounds [115]. Although it is widely thought that *V. cholerae* employs motility and chemotaxis to identify beneficial locations for colonization in the small intestine, the exact role of these mechanisms in infection is still unclear. Host factors like health state or age may affect chemoeffector gradients, thereby attracting the pathogen to unusual sites. Human serum contains a multitude of different amino acids, sugars, and metabolites that could serve as a great source of nutrition and therefore as chemoattractants. Thus, the chemotactic behavior of NOVCs may be one of the key factors facilitating their translocation from the gastrointestinal tract into the bloodstream. Adopting multifaceted strategies for identifying and treating infections like *V. cholerae* bacteremia may therefore require the consideration of the pathogen’s chemotactic behavior.

## 4. Host Immunity and NOVC Bacteremia

Under a normal physiological state, bacteria do not stay long in the bloodstream as they are effectively cleared by circulating immune cells and blood-filtering organs [116]. Notably, the serum complement system and phagocytic leukocytes (neutrophils) are highly effective at eliminating pathogens in the early stages of infection. Although defects in the immune system may afford NOVC the capacity to cause systemic infection, the mechanism by which it evades killing and persists in the bloodstream is largely unknown. Complement-mediated pathogen opsonization and phagocytosis can be inhibited due to polysaccharide capsules made by NOVCs [114]. In Gram-negative bacteria like *V*. *cholerae*, lipopolysaccharide (LPS) is a major molecule that engages host receptors such as the Toll-like receptors (TLRs), nucleotide-binding oligomerization domain (NOD), and leucine-rich repeats (LRR)-containing receptors (NRL), to regulate pro-inflammatory cytokine production. Thus, any structural modification to the LPS may alter the innate immune response of the host. Interestingly, the core LPS of NOVCs does not carry any side chains compared to its toxigenic O1 counterpart, which possesses about 17 side chains of smaller repeated units of 4-NH_2_-4,6-dideoxymannose [117]. Due to the simplified nature of the LPS structure of NOVCs, they may induce restrained immune responses during systemic infection, facilitating their persistence in the bloodstream. This view is consistent with our recent observation where NOVC strains expressing reduced levels of cytokines (IL-1ß and IL-13) showed high serum survival rates [118]. The studies by Bielig et al. also showed that NOVCs can modulate the peptidoglycan content in outer-membrane vesicles (OMVs) to facilitate NLR-mediated immune evasion through quorum sensing regulation [119]. Moreover, it is suggested that decreased expression of the major outer membrane porin (OmpU), which is a target of the natural IgG antibodies for complement-mediated killing [120], may enhance NOVC serum resistance. Serum resistance is also related to the presence of circulating antibodies and more importantly, the serum immunoglobulin A (IgA). IgA has an anti-inflammatory role and can interact with other serum proteins to inhibit the directed chemotaxis of neutrophils for pathogen elimination [121]. The interplay between the innate and adaptive arms of the host defense system may also affect the outcome of NOVC infections. Inadequate complement activation together with reduced CD8^+^ T cells production for the release of potent inflammatory cytokines in immunocompromised individuals [122] possibly contributes to the high burden of *V. cholerae* bacteremia. By secreting proteins that disrupt MAPK signaling, some strains of *V. cholerae* have been shown to impede immune cell-mediated clearance. This prevents the recruitment of host immune components, such as neutrophils and cytokines, which may identify and eliminate the bacteria [123]. It has been reported that multiple natural exposures to *V*. *cholerae* infections over the years lead to protective immunity against the pathogen. Individuals in cholera-endemic settings may have altered their immune systems for increased tolerance to *V*. *cholerae* infections. Therefore, it is probable that the endemicity of diarrheal strains primes the innate immunity of sub-Saharan people, making them less susceptible to invasive bacteremic NOVC strains. Although the concept is not well explored yet, modification in the host immune responses by one pathogen can impact the responses generated against a different or co-infecting pathogen [124]. This hypothesis has been used to explain why individuals in malaria-endemic settings seem to be protected against severe SARS-CoV-2 infections [125,126]. 

## 5. Diagnosis and Treatment of NOVC Bacteremia

Due to the poor prognosis and fatal nature of NOVC bacteremia, early diagnosis is critical for disease management. It is recommended that NOVC should be included in the differential diagnoses of invasive infections [95 ]. A doctor may suspect infection if the patient has recently eaten raw or undercooked seafood, or when a wound infection has developed after exposure to salt or brackish water and showing compatible symptoms. Culture, biochemical tests, PCR amplification, mass spectrometry (MALDI-TOF-MS), and gene sequencing are some of the diagnostic techniques often used in clinical and research laboratories to identify the causative pathogens. During culturing, bacteria appear as large yellow colonies on thiosulfate–citrate–bile salt–sucrose agar, with a typical Gram-negative curved rod morphology [5]. NOVC strains are β-hemolytic on blood agar and oxidase-positive. Aside from being time-consuming and expensive, the facilities required for microbiological culture of blood specimen are not readily available in resource limited settings in Africa. This might lead to delayed diagnosis and underreporting of NOVC bacteremia cases. For molecular confirmation of NOVC, *V. cholerae* gene encoding the outer membrane porin (*ompW*) and the O-antigen *rfb* specific for both O1 and O139 are targeted in Polymerase Chain Reaction (PCR) amplification [127]. NOVC is validated by the presence of *ompW* and the absence of the *ctx* and *tcp* genes. It is worth noting that some NOVC strains might not possess the *OmpW* gene, making whole-genome sequencing the most appropriate method for NOVC identification. Currently, PCR and sequencing-based diagnostic methods are not routinely adapted in most clinics providing primary healthcare, especially in remote areas in Africa. However, a whole-blood or dried blood spot specimen can be taken to centralized laboratories with molecular facilities for analysis. Besides the pathogen, the determination of host responses to infection can give insight into disease diagnosis. Specifically, patterns of host inflammatory mediators such as cytokines and chemokines have been studied as diagnostic markers for NOVC detection [128]. In the case of *V. cholerae* bacteremia, the specific host immune responses remain largely unknown. However, the production of cytokines, including interleukin-1β (IL-1β), IL-6, and IL-17, has been reported to increase in response to acute infections [129]. A recent study also identified the chemokine RANTES, a stably expressed host immune biomarker, as a major immune cell component elicited by *V. cholerae* during bacteremia infection [118]. This suggests that cytokine profiling could be exploited as a specific biomarker for *V. cholerae* bacteremia in addition to the preparation of blood cultures, which currently remains the gold standard for identification. Integrating host and pathogen data into a point-of-care diagnostic test may provide a more accurate diagnosis of *V. cholerae* bacteremia, particularly in sub-Saharan Africa where access to laboratory services is very limited. 

Although no specific treatment guidelines exist for NOVC bacteremia [94], most clinicians have relied on antibiotics. Antibiotic therapy is recommended to be initiated as soon as possible in clinically suspected patients. Several reports have shown significant heterogeneity in the choice of antibiotics, dosage, and treatment duration [7,98]. Cephalosporins (cefotaxime, ceftriaxone, and ceftazidime), beta-lactam/beta-lactamase inhibitors (piperacillin-tazobactam), and fluoroquinolone (ciprofloxacin) are some of the frequently chosen antibiotics in NOVC infection [5]. Intravenous ceftriaxone has been used as part of an empirical parenteral therapy in NOVC bacteremia [7]. In addition, dual-agent therapy (combining a third-generation cephalosporin with tetracycline or fluoroquinolone) is recommended for patients with NOVC septicemia or septic shock [8,15]. Most patients with NOVC infection usually have favorable outcomes after antibiotic therapy. Nevertheless, multi-drug-resistant NOVCs with increasing resistance to first-line antibiotics such as ampicillin and trimethoprim have been reported in both clinical and environmental settings [130], which could potentially influence the duration and success of therapy. Moreover, the use of antibiotic therapy alone was ineffective for certain individuals who could not recover and eventually lost their lives due to complicated underlying diseases [82,83]. Thus, it is recommended that antimicrobial treatment in patients with NOVC bacteremia must be based on the patient’s clinical history and antibiogram data of the cultured isolate. However, contrary to toxigenic *V. cholerae*, antimicrobial resistance patterns for NOVC bacteremia isolates are rarely published. A recent study indicated greater susceptibility to sulfonamides, quinolones, aminoglycosides and other antibiotics for a bacteremia strain isolated from a 58-year-old male Chinese patient with cirrhosis [102]. Another study in Columbia involving a 79-year-old woman with NOVC bacteremia showed sensitivity to penicillins, cephalosporins, carbapenems, aminoglycosides, quinolones, and trimethoprim/sulfamethoxazole, with intermediate sensitivity to tetracyclines [9]. Xiang et al. [8] also mentioned that some isolates causing NOVC bacteremia in an inland city in China were sensitive to commonly used antibiotics including second and third generation cephalosporins such as ceftriaxone, ceftazidime and cefoxitin. Taken together, these data indeed suggest that NOVC bacteremia is treatable once the susceptibility of the causative pathogen is established. 

## 6. Conclusions

Although *V. cholerae* bacteremia appears to be rare in Africa, the disease outcome can be fatal. In a moment of crisis, sub-Saharan Africa’s current laboratory and healthcare infrastructures are largely insufficient to address its diagnostic challenges. Invasive bacterial infections may be an underappreciated cause of death in Africa. The lack of laboratory expertise and minimal awareness among clinicians may contribute to the underreporting of NOVC cases in Africa. Considering the gradual increase of *V. cholerae* bacteremia across Asia, Europe, and America in recent times, active surveillance and much research attention for this neglected infection is highly recommended in Africa. Putting into practice the strategies that the WHO has recommended to support surveillance of newly emerging and re-emerging illnesses would significantly reduce the threat that unpredictable pathogens like NOVC present to public health. To identify cases that fit the predetermined case criteria of NOVC bacteremia, active surveillance will entail proactive interaction with the healthcare system by leveraging medical facilities to promote follow-up and evaluation of clinical data. In Africa, where rigorous and coordinated infectious disease surveillance systems are somewhat lacking, the heightened screening and reporting of NOVC should be strongly encouraged. Ongoing cholera disease surveillance programs in Africa should be expanded to include all vibrios just like the COVIS system in the United States. Such an approach may not only improve our understanding of NOVC epidemiology within the African landscape but also guide the diagnosis and establish adequate therapeutic strategies for these pathogens.

## 7. Highlights

The low reported incidence of *V. cholerae* bacteremia infection in Africa is peculiar as cholera infections are endemic in Africa;The discrepancy between the global increase of *V. cholerae* bacteremia and low reported incidences in Africa may be explained by qualitatively and quantitatively insufficient routine laboratory diagnosis, which could have fatal patient outcomes;Rapid point-of-care diagnostic tools integrating information from both host responses and *V. cholerae* detection in blood are highly recommended;Advocacy to create awareness of NOVC bacteremia among African clinicians is paramount.

## Figures and Tables

**Figure 1 tropicalmed-09-00103-f001:**
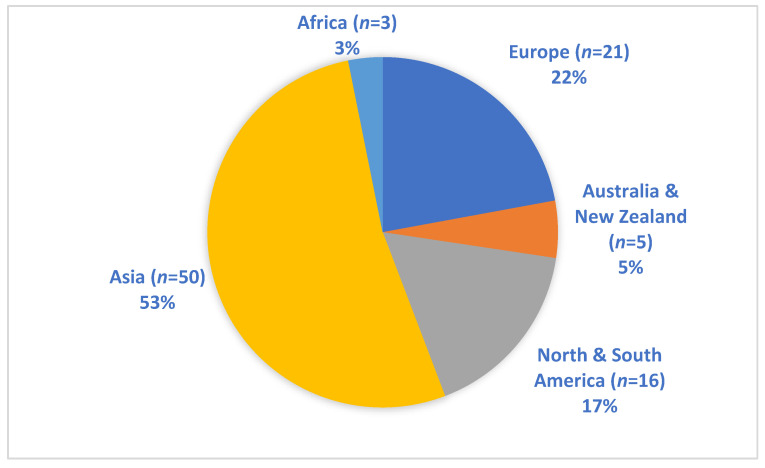
Global distribution of *V. cholerae* bacteremia cases reported and published between the years 2000–2023.

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
