# Peer review of "Vibrio cholerae Bacteremia: An Enigma in Cholera-Endemic African Countries"

_tropicalmed, 2024, doi:10.3390/tropicalmed9050103_

Round 1
Reviewer 1 Report
Comments and Suggestions for Authors
This is a well-written review of what is currently known about the epidemiology, host risk factors, pathogenesis, and diagnostics of Vibrio cholerae bacteremia, a syndrome distinct from the more heralded diarrheal syndrome "cholera." The authors summarize current knowledge of the causative pathogens, the non-toxigenic (non-O1/O139) strains of V. cholerae known as NOVC. They pay particular attention to the discrepant incidence of bacteremia in sub-Saharan Africa, in light of the endemicity of V. cholerae and high rates of diarrheal illness there. Namely, while global rates of V. cholerae bacteremia are increasing, this is not occurring at the same rate in sub-Saharan Africa, where the authors hypothesize there should be more bacteremia. They postulate several mechanisms for this discordance, including virulence factor differences and the absent or limited infrastructure to diagnose bacteremia, which is likely a key factor. The authors appropriately conclude that increased surveillance of V. cholerae bacteremia in Africa may have significant clinical implications.
One question that came to mind was whether the endemicity of diarrheal strains primes the innate immunity of sub-Saharan people, making them less susceptible to invasive / bacteremic NOVC strains. I don't think this was discussed in the manuscript but would be worth exploring.
There are minor grammatical / syntax errors, that should be edited for clarity. For example, section "Pathogenesis of NOVC bacteremia" could be improved for clarity given the numerous virulence factors that are discussed - and whether they are present in NOVC and epidemic V. cholerae - whether the evidence supports their importance in bacteremia versus non-invasive diseases - should be made explicit.
Overall, a very interesting review whose publication I would support.
Comments on the Quality of English LanguageThere are minor grammatical / syntax errors, that should be edited for clarity. For example, section "Pathogenesis of NOVC bacteremia" could be improved for clarity given the numerous virulence factors that are discussed - and whether they are present in NOVC and epidemic V. cholerae - whether the evidence supports their importance in bacteremia versus non-invasive diseases - should be made explicit.
Author Response
Q1. One question that came to mind was whether the endemicity of diarrheal strains primes the innate immunity of sub-Saharan people, making them less susceptible to invasive / bacteremic NOVC strains. I don't think this was discussed in the manuscript but would be worth exploring.
Response:
Thank you for the excellent suggestion. We have taken note of the recommendation and discussed how multiple natural exposures to V. cholerae infections over the years may lead to protective immunity and increased tolerance for people living in cholera endemic regions. Also, added the suggestion where endemicity of diarrheal strains primes the innate immunity of sub-Saharan people, making them less susceptible to bacteremic NOVC strains. These changes can be found under the 'Host immunity and NOVC bacteremia' section (Lines 203-212).
Q2. There are minor grammatical / syntax errors, that should be edited for clarity. For example, section "Pathogenesis of NOVC bacteremia" could be improved for clarity given the numerous virulence factors that are discussed - and whether they are present in NOVC and epidemic V. cholerae - whether the evidence supports their importance in bacteremia versus non-invasive diseases - should be made explicit.
Response:
We appreciate this comment. Major changes have been made to improve clarity of the pathogenesis section, highlighting the contribution of some selected virulence factors to Vibrio cholerae bacteremia (Lines 144-146).
Reviewer 2 Report
Comments and Suggestions for Authors
The manuscript "Vibrio cholerae bacteremia: An enigma in cholera endemic African countries" examines non-toxigenic strains of V. cholerae in Africa. While the topic is interesting, the manuscript primarily offers general information on non-O1/non-O139 V. cholerae (NOVC), which has been extensively reviewed previously (e.g. 10.1016/j.meegid.2021.104726). The analysis of specific cases traced to Africa is relatively brief.
Major comments:
1. The distribution of V. cholerae bacteremic cases, as presented in Figure 1, lacks information on the total number of cases in the regions.
2. The primary conclusion of the manuscript suggests that rare case reports may be attributable to diagnostic challenges in Africa. To support this assertion, a comparison with other diseases exhibiting similar trends should be included.
3. A detailed analysis of the isolated strains in the reported cases should be provided.
Author Response
Major comments:
1. The distribution of V. cholerae bacteremic cases, as presented in Figure 1, lacks information on the total number of cases in the regions.
Response:
Information on the total number of published reports for V. cholerae bacteremic cases in the regions is now provided and presented in Figure 1.
2. The primary conclusion of the manuscript suggests that rare case reports may be attributable to diagnostic challenges in Africa. To support this assertion, a comparison with other diseases exhibiting similar trends should be included.
Response:
This is a very useful suggestion, but we think it is outside the remit of this review. A typical example for such trend can be seen with malaria. In Africa, 4 species of Plasmodium parasites lead to human disease with P. falciparum being the most prevalent and causing 99.7% of the infections. Diagnosis of non-falciparum infections are difficult and often missed, especially as current malaria RDTs do not reliably or specifically detect the other species, underestimating their prevalence and disease severity. Individuals with non-falciparum infections may continue to transmit parasites within their communities if left untreated. The non-falciparum parasites can cause severe malaria just as P. falciparum.
3. A detailed analysis of the isolated strains in the reported cases should be provided.
Response:
We have expanded on the specific cases traced to Africa and provided some details to the isolated strains from the patients (Lines 49-59).
Reviewer 3 Report
Comments and Suggestions for Authors
In the commentary “ Vibrio cholerae bacteremia: An enigma in cholera endemic African countries”, the authors discuss and analyse the low incidence of Vibrio cholerae bacteremia, even though the pathogen is endemic in the region.
Strengths:
- The commentary tackles a crucial and under-investigated aspect of cholera in endemic African regions.
- The analysis of low bacteremia incidence despite endemicity is insightful and raises valuable questions.
- The writing is clear, concise, and well-organized.
Areas for Improvement:
1. Clarification of Reported Cases:
- Figure 1: While the figure is visually appealing, it's crucial to explicitly state that it depicts reported and published cases, not the overall number of occurrences. This distinction is vital for accurate interpretation of the data and understanding true prevalence.
- Consider differentiating between confirmed and probable cases, if possible, to provide further clarity.
2. Data Presentation:
- While the narrative presentation of data is clear, presenting some key findings in a table would enhance clarity and accessibility.
- A table could effectively show:
- Progression of reported cases over time.
- Geographical distribution of reported cases.
- Comparison of reported cases across different countries or regions.
3. Antibiotic Susceptibility Data:
- If available, even a brief mention of antibiotic susceptibility profiles for reported bacteremia strains would be valuable. This information could provide crucial context regarding potential treatment challenges and inform future research directions.
- If such data is unavailable, acknowledging its absence and highlighting its importance for future studies would be beneficial.
Overall, this commentary presents valuable insights and raises important questions. Implementing the suggested improvements would further strengthen its impact and contribute significantly to the understanding of V. cholera bacteremia the African regions.
Comments on the Quality of English LanguageN/A
Author Response
Areas for Improvement:
- Clarification of Reported Cases:
- Figure 1: While the figure is visually appealing, it's crucial to explicitly state that it depicts reported and published cases, not the overall number of occurrences. This distinction is vital for accurate interpretation of the data and understanding true prevalence.
- Consider differentiating between confirmed and probable cases, if possible, to provide further clarity.
Response:
We really appreciate the comment. The legend to figure 1 explicitly states the global reported and published cases of V. cholerae bacteremia. In nearly all cases, the causative pathogen(s) was identified and confirmed.
2. Data Presentation:
- While the narrative presentation of data is clear, presenting some key findings in a table would enhance clarity and accessibility.
- A table could effectively show:
- Progression of reported cases over time.
- Geographical distribution of reported cases
- Comparison of reported cases across different countries or regions.
Response:
In addition to figure 1, we have provided a table (Table 1) to show the progression of reported cases in 5-year intervals from 2000 to 2023 across different countries.
3. Antibiotic Susceptibility Data:
- If available, even a brief mention of antibiotic susceptibility profiles for reported bacteremia strains would be valuable. This information could provide crucial context regarding potential treatment challenges and inform future research directions.
- If such data is unavailable, acknowledging its absence and highlighting its importance for future studies would be beneficial.
Response:
Thank you for the excellent suggestion. We have taken note of the recommendation and updated the "Diagnosis and treatment of NOVC bacteremia" with more information on antibiotic susceptibility profiles for reported bacteremia strains (Lines 268-279).
Round 2
Reviewer 2 Report
Comments and Suggestions for Authors
The revised version of the review contains more information on the cases of NOVC in Africa, it might be suitable for publication in Tropical Medicine and Infectious Disease.